# Quantifying the drivers and predictability of seasonal changes in African fire

Yan Yu [1,7], Jiafu Mao [2,7 ✉], Peter E. Thornton [2], Michael Notaro[3], Stan D. Wullschleger[2], Xiaoying Shi[2], Forrest M. Hoffman [4,5] & Yaoping Wang [6]

Africa contains some of the most vulnerable ecosystems to fires. Successful seasonal prediction of fire activity over these fire-prone regions remains a challenge and relies heavily on in-depth understanding of various driving mechanisms underlying fire evolution. Here, we assess the seasonal environmental drivers and predictability of African fire using the analytical framework of Stepwise Generalized Equilibrium Feedback Assessment (SGEFA) and machine learning techniques (MLTs). The impacts of sea-surface temperature, soil moisture, and leaf area index are quantified and found to dominate the fire seasonal variability by regulating regional burning condition and fuel supply. Compared with previously-identified atmospheric and socioeconomic predictors, these slowly evolving oceanic and terrestrial predictors are further identified to determine the seasonal predictability of fire activity in Africa. Our combined SGEFA-MLT approach achieves skillful prediction of African fire one month in advance and can be generalized to provide seasonal estimates of regional and global fire risk.

[1] Atmospheric and Oceanic Sciences Program, Princeton University, Princeton, NJ, USA. [2] Oak Ridge National Laboratory, Environmental Sciences Division and Climate Change Science Institute, Oak Ridge, TN, USA. [3] Nelson Institute Center for Climatic Research, University of Wisconsin-Madison, Madison, WI, USA. [4] Oak Ridge National Laboratory, Computational Sciences and Engineering Division and Climate Change Science Institute, Oak Ridge, TN, USA. [5] Department of Civil & Environmental Engineering, University of Tennessee, Knoxville, TN, USA. [6] Institute for a Secure & Sustainable Environment, University of Tennessee, Knoxville, TN, USA. [7] These authors contributed equally: Yan Yu, Jiafu Mao. ✉email: maoj@ornl.gov

Fires affect ecosystems and atmospheric composition, as well as human infrastructure and safety[1]. In recent decades, about 50% of fire-related carbon emissions and ~70% of global burned areas occurred across African subtropical savannah systems, both north and south of the equator[2,3] (Supplementary Fig. 1). African fires mediate the interannual variability in atmospheric concentrations of $CO_2$ and other greenhouse gases[4]. In sub-Saharan Africa, biomass burning also plays a key role in regional climate and the hydrological cycle through the emission of dust and black carbon aerosols that affect the energy and water cycles[5]. For example, an atmospheric model simulation showed that biomass burning aerosols emitted from southern African fires caused surface cooling and inhibited convective rainfall[6]. In addition, fires threaten human societies in Africa by emitting large amounts of atmospheric pollutants that are harmful to both property and the population[1]. A deeper understanding of the major environmental drivers behind African fire evolution would aid in the capacity to accurately predict seasonal fire activity, thus fostering better fire management practices in these vulnerable ecoregions[7].

Fire activity across Africa is highly variable and has been previously attributed to changes in weather patterns, shifting plant communities, and human activity[3,5,8–11]. Variation in the extent of fire burned area is widely believed to be dependent on vegetation composition and distribution[1], as well as air and soil controls on fuel drying[12]. Satellite observations have indicated a decline in burned area across northern Africa since 1998 (ref. [13]). Contributions of demographic and socioeconomic changes, such as population growth and cropland expansion, have been comparable to climatic and ecological factors in driving the observed decline in African burned area[3,13,14]. Prior knowledge of natural and anthropogenic drivers of Africa's fires serves as a theoretical basis for building predictive models of fire activity. However, seasonal prediction of fire changes must address additional challenges, given that relevant environmental and socioeconomic states have limited predictability on the seasonal timescale[15]. For example, seasonal climate forecasts used to predict fire activity resulted in insignificant correlations between observed and predicted burned areas across the majority of Africa[16].

Observed variations in regional climate and hydrology across sub-Saharan Africa have been shown to be sensitive to global sea-surface temperature (SST) variability and regional land surface changes on the seasonal timescale[17,18]. Such oceanic and terrestrial controls on African climate variability can potentially enhance the ability to predict African fire at relevant timescales, since the oceanic and terrestrial states generally exhibit longer memory than does the atmosphere[19]. However, neither specific oceanic and terrestrial drivers nor their individual contributions to the seasonal predictability of African fire activity have been sufficiently explored. Past studies of oceanic drivers of African fire have focused mainly on the El Niño-Southern Oscillation (ENSO), identifying spatio-temporally heterogeneous responses in precipitation and the resulting fire activity[3,20]. Limited effort has been devoted to addressing other oceanic drivers, such as tropical Atlantic SSTs[21], that significantly influence African regional climate. Moreover, while different modes of variability in SSTs presumably co-impact regional fire activity in Africa[22], their synergistic and independent roles lack systematic exploration in either observations or model simulations. Vegetation indices, such as leaf area index (LAI), may influence African fire through multiple mechanisms. Expanded vegetation cover provides additional fuel for burning, particularly in semi-arid landscapes[1]. Enhanced vegetation growth, however, alters the energy, moisture, and momentum fields, and causes wetter, cooler, and less windy conditions in a portion of sub-Saharan Africa[17,18], potentially inhibiting the biomass burning. Increased soil moisture content is believed to inhibit biomass burning[23]. While wetter soils may further strengthen vegetation growth, intensifying biomass burning through greater accumulation of aboveground vegetation and litter fuels. Although the vegetation and soil components are parameterized in the current generation of fire models[23], their linear and nonlinear impacts on fire activity remain relatively simplified, especially on the seasonal timescale.

Main challenges to extracting key oceanic and terrestrial drivers of African regional fire activity include the following: (1) impacts of fire on vegetation typically outweigh the feedbacks from vegetation to fire, and (2) oceanic and land surface anomalies are usually intercorrelated, so longer data records are required for reliable statistical separation of their individual influences. The Stepwise Generalized Equilibrium Feedback Assessment (SGEFA), a lagged covariance statistical method, was developed to simultaneously assess the impacts of oceanic and terrestrial drivers on regional climate[24] on the basis of differentiated memory of forcing and response variables. The ability of SGEFA to separate linear contributions of intercorrelated oceanic, vegetation, and soil moisture forcings on seasonal timescale has been rigorously demonstrated by dynamic experiments and observational applications[17,18,25,26]. However, to build a more comprehensive fire prediction model, machine learning techniques (MLTs) are particularly useful because they provide unique tools to investigate the nonlinear and complex effects of natural and anthropogenic factors on fire activity[12,22,23]. MLTs have been extensively used for seasonal forecasting and long-term projection of geoscience variables, but require process-based guidance for predictors selected from a high-dimensional data pool[27]. In addition, to avoid possible overfitting using limited observational records, MLTs need to be informed by pre-identified key environmental drivers of fire.

Motivated by gaps in our knowledge of ocean–land feedbacks to African fire, this study aims to robustly characterize oceanic and terrestrial drivers and their contribution to the predictability of fire activity in Africa, with a focus on the seasonal timescale. Here, SGEFA is applied to quantify dominant oceanic and terrestrial factors affecting seasonal variability of African fire. Guided by SGEFA, MLTs are subsequently used to develop a seasonal fire prediction system across sub-Saharan Africa. Based on the combined SGEFA-MLT analytical framework (Supplementary Fig. 2 and see the Methods section), the present research demonstrates that seasonal changes of African fire carbon emissions and burned area fraction are primarily sensitive to variations in SST, LAI, and soil moisture, which are mechanistically connected to the regional burning conditions and biomass fuel supply. Benefiting from the inclusion of SGEFA-identified, slowly evolving oceanic and terrestrial predictors, the MLT-based approach effectively predicts African fire activity 1 month in advance.

## Results

**Oceanic and terrestrial drivers of African fire variability.** According to SGEFA, fire activity across both the northern and southern arid/semi-arid African ecoregions (Supplementary Fig. 1) exhibits significant sensitivity to variability in SSTs, LAI, and soil moisture during the dry fire-active season (Fig. 1). Northern Africa's fire is sensitive to variability in tropical ocean SSTs, especially those from the tropical Atlantic Ocean during the dry season (November to March) and the tropical Indian Ocean during the wet season (April to September). During the boreal winter, North Atlantic Ocean SSTs play a substantial role in regulating the northern African fire carbon emissions (Fig. 1a). SSTs from Southern Hemispheric oceans, especially the South Atlantic Ocean, exert relatively strong control compared with the

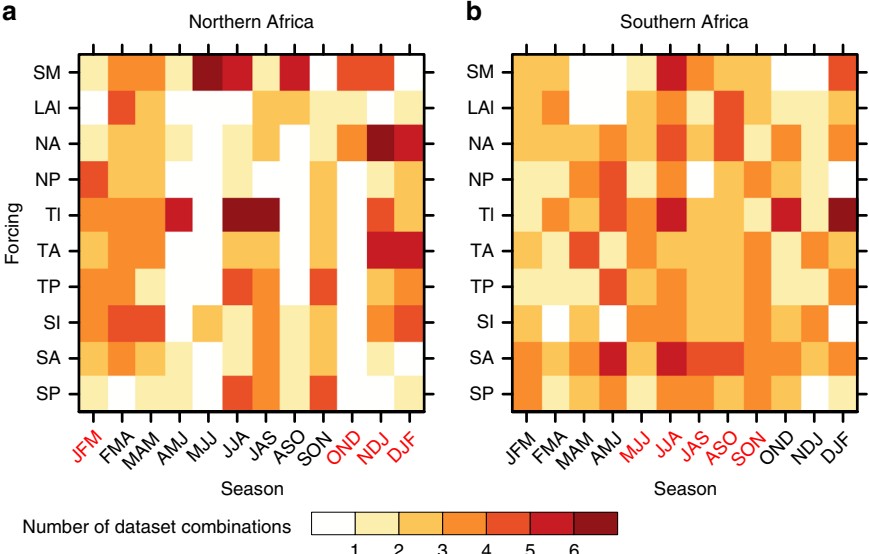

**Fig. 1 Robustness of controls on African fire by season.** The multi-dataset robustness of oceanic and terrestrial controls on African fire carbon emission is assessed by the Stepwise Generalized Equilibrium Feedback Assessment (SGEFA) by season. Color represents the number of dataset combinations [out of the currently examined six dataset combinations (Supplementary Table 1)] that indicate significant ($p < 0.1$) response of regional average fire carbon emission to either of the leading two principal components' (PCs) time series corresponding to the sea-surface temperature (SST) empirical orthogonal functions (EOFs) from eight oceanic basins [North Atlantic (NA), North Pacific (NP), tropical Indian (TI), tropical Atlantic (TA), tropical Pacific (TP), South Indian (SI), South Atlantic (SA), and South Pacific (SP)], and regional average leaf area index (LAI) and 0–10 cm soil moisture (SM), across **a** northern Africa and **b** southern Africa. In both **a**, **b**, labels on the x-axis stand for 3-month seasons, for example, January–March (JFM) for January, February, and March. The fire-active season is highlighted with red x-axis labels. The forcings are detailed in the Methods section.

tropical and Northern Hemispheric oceans on southern African regional fire during the fire-active season (May to November) (Fig. 1b). The mechanisms underlying key oceanic drivers of African fire variability, namely, ENSO and the Atlantic Niño mode, are explored in detail in the next subsection. In terms of terrestrial drivers of African fire, the observed effects from soil moisture changes are more robust than LAI changes, especially during the wet season in northern Africa (Fig. 1a). The mechanistic controls of the terrestrial drivers of African fire variability are explored in detail in the next subsection.

The SGEFA-based observational analysis identifies the spatially heterogeneous influence of ENSO and relatively homogeneous influence of the Atlantic Niño mode on African fire activity, with the response magnitude of both oceanic drivers peaking in the fire-active season (Fig. 2a, b, e, f, i, j and Supplementary Fig. 4). Across a large portion of northern Africa, especially the northern Sahel and West African monsoon regions, El Niño (inlet in Fig. 2e) typically supports enhanced fire carbon emissions and expanded burned area during the fire-active season in boreal winter (Fig. 2a, e and Supplementary Fig. 4). The increased fire activity in northern and western Sahel is largely associated with El Niño-induced anomalous low-level warming and consequential fuel drying (Supplementary Fig. 5a, e). Similar meteorological responses to El Niño (Supplementary Fig. 5a, e) also support enhanced fire emissions and expanded burned area across southwestern Africa during the fire-active season in boreal summer (Fig. 2e, i and Supplementary Fig. 4). The SST anomalies associated with positive Atlantic Niño (inlet in Fig. 2f) are responsible for positive fire anomalies across most of Africa (Fig. 2b, f, j and Supplementary Fig. 4). Anomalously warm SSTs over the tropical Atlantic Ocean, with greater warming to the south of the Equator, support anomalous warming over southern Africa and drying over northern Africa (Supplementary Fig. 5b, f), thereby enhancing fire activity across the majority of Africa during the fire-active season. The season-dependent response in fire to oceanic drivers (Fig. 2a, b, i, j) partly leads to the spatial

heterogeneity of the seasonal maximum-magnitude responses presented in Fig. 2e–f and Supplementary Fig. 4.

The SGEFA analysis uncovers generally negative responses in African fire to positive anomalies in soil moisture and complex responses to LAI changes, with greater response magnitudes to soil moisture anomalies during the fire-active season (Fig. 2c, g, k, d, h, l and Supplementary Fig. 4). Across most of Africa, wetter soils inhibit biomass burning (Fig. 2d, h, l and Supplementary Fig. 4) through surface low-level cooling and elevated amounts of precipitation (Supplementary Fig. 5d, h). However, the influence of LAI on fire activity in Africa is spatially heterogeneous (Fig. 2g and Supplementary Fig. 4). Positive anomalies in LAI indicate a higher amount of available fuel for biomass burning, leading to enhanced fire activity over portions of the West African monsoon region and grasslands in southern Africa (Fig. 2g and Supplementary Fig. 4). Similarly, positive anomalies in LAI cause an overall increase in North African regional average fire emission and burned area fraction during the fire-active season in boreal winter and spring (Fig. 2c and Supplementary Fig. 4). However, surface cooling and decreased near-surface wind speed associated with positive LAI anomalies (Supplementary Fig. 5c, k) provide unfavorable meteorological conditions for biomass burning, thereby inhibiting fire activity across the majority of southern Africa during the fire-active season in boreal summer (Fig. 2g, k and Supplementary Fig. 4).

**Seasonal predictability of African fire.** The predictability of fire anomalies across both northern and southern African ecoregions is substantially enhanced by including the slowly evolving oceanic and terrestrial forces quantified by SGEFA (Fig. 3 and Supplementary Fig. 7). Optimal fire predictability is represented by the ensemble-mean-squared correlation coefficient ($R^2$) between the observed and predicted time series produced by the best MLT using all predictors, and reflects the all-season average of fire predictability. It decreases by a lead time from 0.60 (0.52–0.77

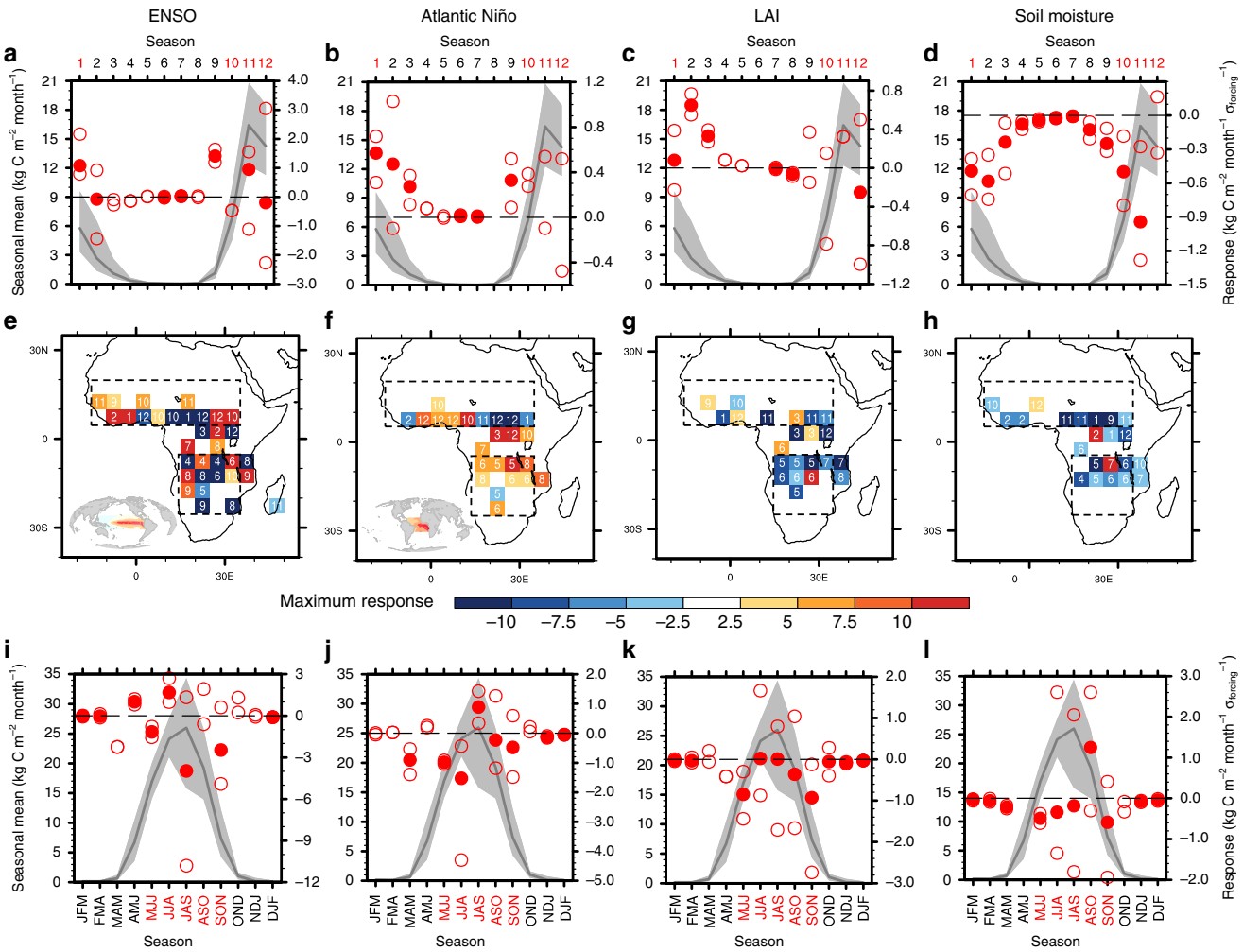

**Fig. 2 African fire response to key environmental drivers.** Response of African fire carbon emissions to the most important environmental forcings is assessed by the Stepwise Generalized Equilibrium Feedback Assessment (SGEFA). These forcings include **a**, **e**, **i** El Niño-Southern Oscillation (ENSO), **b**, **f**, **j** Atlantic Niño mode, **c**, **g**, **k** leaf area index (LAI), and **d**, **h**, **l** soil moisture. Seasonal cycle of the **a–d** northern and **i–l** southern African regional average climatology, with line and shading representing mean and interannual standard deviation, referring to the left $y$-axis (unit: kg C m$^{-2}$ month$^{-1}$), and response to the corresponding forcing, with the filled and open circles indicating the multi-dataset average and 10th and 90th percentiles, referring to the right $y$-axis (unit: kg C m$^{-2}$ month$^{-1}$ $\sigma_{forcing}^{-1}$). A missing filled circle in **a–d** and **i–l** indicates insignificant multi-dataset average response to the specific forcing. **e–h** Spatial distribution of the season (number representing each 3-month season, e.g., 1 for January–March) and sign and magnitude (color) of the maximum absolute response to the corresponding forcing. The spatial patterns of the ENSO and Atlantic Niño forcing are demonstrated in the inserted global map in **e** and **f**, respectively. In these inlets, positive and negative anomalies in sea-surface temperature are represented by red and blue colors, respectively. The boxes in **e–h** indicate the geographic location of the currently assessed northern and southern African ecoregions. Only statistically significant responses ($p < 0.1$) are shown here. In **i–l**, labels on the $x$-axis stand for 3-month seasons, e.g., JFM for January, February, and March. The fire-active season is highlighted with red labels on the $x$-axis.

across ensemble members) at 1 month in advance to 0.16 (0.07–0.18) at 6 months in advance. Fire carbon emission in northern Africa can be accurately predicted at 1 month in advance if only using the selected oceanic and terrestrial predictors, with an ensemble mean $R^2$ of 0.51 (0.39–0.61), compared with $R^2$ of 0.38 (0.25–0.41) when using the atmospheric and socioeconomic predictors identified by previous studies (Supplementary Table 1). Furthermore, season-specific models that are built and applied by season always perform better than the all-season models, suggesting season-dependent environmental controls on African fire activity, as confirmed previously. Based on the current analytical framework, the northern Africa ecoregion generally exhibits higher predictability of its fire activity than southern Africa, indicating a higher dependency of northern African fire activity on the selected environmental and socioeconomic factors and/or a higher predictability with these factors.

The predictability of African fire anomalies and the contribution of oceanic and terrestrial predictors vary by season, especially across the northern African ecoregion (Fig. 4 and Supplementary Fig. 8). In northern Africa, the ensemble-mean overall predictability ($R^2$) of fire carbon emission varies from 0.78 in March–May to 0.23 in August–October (ASO). Southern Africa shows smaller inter-season variability in fire predictability, varying from 0.51 in April–June to 0.22 in February–April (FMA). Attributed to longer persistence time of the anomalies, oceanic and terrestrial predictors demonstrate higher prediction skills than the combined atmospheric and anthropogenic factors in predicting fire carbon emission in both African ecoregions for all seasons, with the highest annual-average importance scores assigned to soil moisture (Supplementary Fig. 6). Moreover, using only the oceanic and terrestrial drivers, the predictability of fire carbon emission in northern Africa during the fire-active season

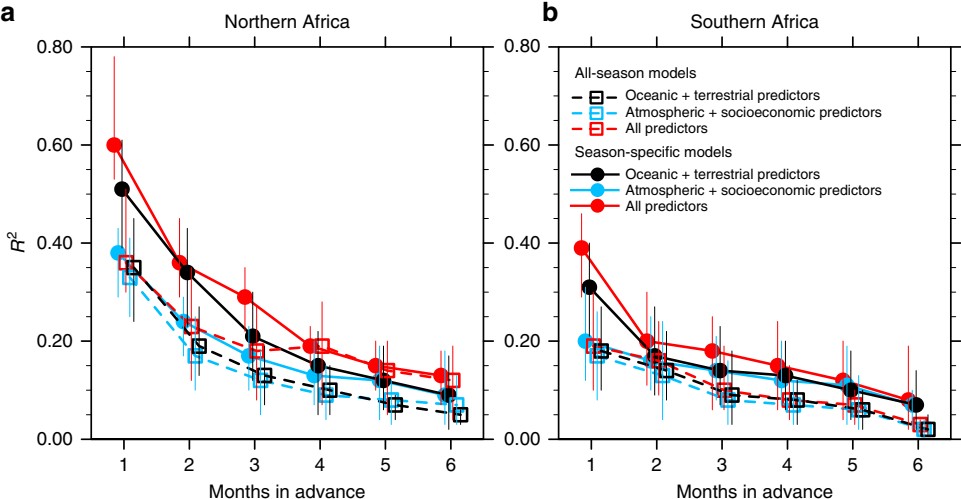

**Fig. 3 Predictability of African fire by leading time.** The predictability of African fire carbon emission anomalies is estimated using multiple machine learning techniques (MLTs) as a function of lead time. The predictability is represented by the squared correlation coefficient ($R^2$) between the predicted and observed monthly anomalies ($n = 60$) of the regional average fire carbon emissions across the **a** northern and **b** southern African ecoregions. The assessed sets of predictors include previously identified atmospheric and socioeconomic predictors (blue), currently identified oceanic and terrestrial predictors (black), and the combination of all these predictors (red). The assessed models include season-specific models (filled circles), in which the MLTs are built and applied by season, and all-season models (open squares). The circles and squares indicate the mean $R^2$ across the 100 ensemble members of the best MLT (see the Methods section), and the vertical lines indicate the range of 10th and 90th percentiles of the 100 ensemble members.

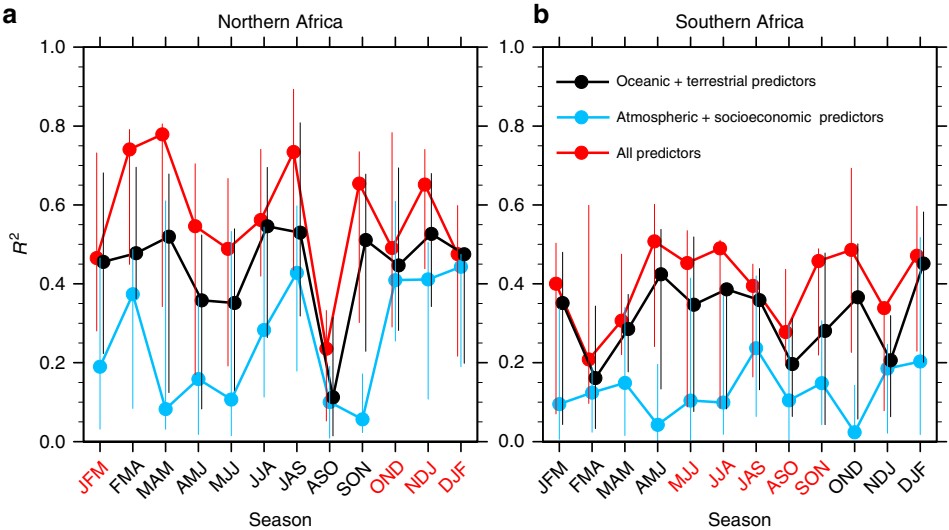

**Fig. 4 Predictability of African fire anomalies by season.** The predictability of regional average fire carbon emission anomalies in each season is estimated using multiple machine learning techniques (MLTs). The predictability is represented by the squared correlation coefficient ($R^2$) between the predicted and the observed monthly anomalies ($n = 15$, see the Methods section) of the regional average fire carbon emission across the **a** northern and **b** southern African ecoregions. The assessed sets of predictors include previously identified atmospheric and socioeconomic predictors (blue), currently identified oceanic and terrestrial predictors (black), and the combination of all these predictors (red). Season-specific prediction models are assessed here at 1 month in advance. The filled circles represent the mean $R^2$ across the 100 ensemble members of the best MLT, and the vertical indicates the range of 10th and 90th percentiles of the 100 ensemble members. The fire-active season is highlighted with red labels on the $x$-axis.

is close to the overall predictability derived from all the environmental and socioeconomic factors. However, in ASO, northern Africa shows relatively weak fire predictability, likely caused by reduced impacts of broad-scale atmospheric circulation from key ocean-atmosphere teleconnection patterns[28], including ENSO and Atlantic Niño mode (Figs. 1a and 2a, b).

## Discussion

This study applies the combined SGEFA-MLT analytical framework that benefits from the capabilities of both methodologies.

Although MLTs have been widely applied for disentangling the controls and building prediction models of regional and global fire activity[23,29,30], they have been criticized as being black boxes and are seldom considered optimal for examining underlying mechanisms and processes[23]. Furthermore, given the limited availability of African fire records and the small training dataset provided, the predictors chosen to be included in MLT models need process-based guidance on the drivers of fire variability. Therefore, the successful application of MLTs largely benefits from the pre-identification of key oceanic and terrestrial drivers

using SGEFA. On the other hand, due to the required long memory of forcing variables, SGEFA is limited in assessing the contribution of other atmospheric and anthropogenic regulators to seasonal fire changes. MLTs, however, are adequate for separating the contribution of various sources and thereby provide full rank of their importance (Supplementary Fig. 6). By combining the SGEFA with MLTs, this study thus presents a promising analytical framework for investigating both the driving mechanisms and predictability of regional and even global fire variability. Nevertheless, efficient application of the SGEFA-MLT framework onto other fire-active regions would require careful selection of oceanic, terrestrial, atmospheric, and socioeconomic predictors that are specifically relevant to the focal region. For example, the potential influences of Madden Julian Oscillation[31] and broader-scale vegetation anomalies[32] need to be fully assessed when investigating the drivers and predictability of fire variability in Australia.

The SGEFA analysis identifies spatially heterogeneous and seasonally dependent influences of key oceanic and terrestrial forces on African fire activity. The currently identified, spatially heterogeneous ENSO signal on the African fire carbon emission and burned area fraction is largely consistent with previous conclusions based on regression and correlation[3,20]. Beyond ENSO, this study further demonstrates the observed role of tropical Atlantic Ocean SSTs on African fire activity, through the influence of SSTs on the climate of Sahel and West African monsoon regions as noted in previous studies[33]. Compared to LAI, the observational analysis further demonstrates more robust effects on fire from changes in soil moisture. This stronger role of soil moisture on African fire activity is supported by previous modeling and observational studies[34]. They concluded that the Sahel ecoregion represents one of the global hotspots of soil moisture–atmosphere coupling, with direct soil moisture feedbacks outweighing the influence of vegetation[18].

The current analytical framework will aid the development of process-oriented fire simulation and prediction for offline or online fire models, such as those participating in the Fire Modeling Intercomparison Project (FireMIP)[35]. Previous studies have demonstrated that the majority of FireMIP models were able to capture the spatial mean characteristics of observed global fire, but failed to reproduce the seasonal to interannual variations[13,36]. Beyond the already identified incorrect response to land use and land cover changes[36], potential misrepresentation of season-dependent, primary environmental drivers may induce unrealistic simulation of regional fire variability. Therefore, the SGEFA-based process quantification would provide a valuable analytical framework for deriving relationship metrics for comprehensive benchmarking of fire models.

Uncertainty in present characterization of environmental contributions to African fire variability and predictability is primarily caused by limitations in observational data availability. For example, the low importance of lightning as a predictor of fire emission (Supplementary Fig. 6) is likely attributed to the dominant role of agricultural practice on fire ignition in Africa, but it might also be an artifact of the lightning data quality and temporal coverage. The current analytical framework led to the finding that the contribution of socioeconomic predictors to the predictability of African fire is limited to population density and land cover change (Supplementary Fig. 6). However, insufficient social and economic observations as well as limited temporal resolution and untested data quality likely resulted in underestimation of human contributions to the African fire predictability. This is particularly critical given the increasing role of the effects of humans on fire behavior across Africa[3,13]. In addition, the predictability of African fire activity is based on a limited number of MLTs trained with relatively short datasets and small

sets of predictors, likely inducing uncertainty in the fire predictability (e.g., vertical lines in Figs. 3 and 4). Furthermore, the current prediction effort using MLTs is mainly focused on area-averaged fire activities in broad African ecoregions (e.g., the northern and southern arid/semi-arid areas denoted in Supplementary Fig. 1). Such summaries potentially oversimplify the drivers and predictability of fire variability across local landscapes. Overcoming these limitations to achieve SGEFA-MLT application at fine scales would require longer observational data records with higher spatio-temporal resolution, and increased numbers of observable environmental variables.

In summary, we have developed and applied a combined SGEFA-MLT analytical tool to quantify the seasonal drivers and predictability of African fire variability. Based on SGEFA, the seasonal variability of fire carbon emissions and burned area fraction in both the northern and southern Africa ecoregions are characterized as being primarily sensitive to changes in SST, LAI, and soil moisture, which alter the regional burning conditions and biomass fuel supply. The MLT-based seasonal predictability of fire activity over two selected African ecoregions benefits substantially from the inclusion of SGEFA-identified, slowly evolving oceanic and terrestrial predictors. By using the SGEFA-selected environmental variables and other atmospheric and anthropogenic drivers identified in the literature, the seasonal anomalies of African fire are successfully predicted 1 month in advance, outperforming previous fire forecasts mainly based on seasonal climate forecasts[37]. The regional diagnostic and prediction framework provides an encouraging basis for building a global fire early warning system.

## Methods

**SGEFA-based quantification of environmental controls**. The multivariate statistical approach, SGEFA, assesses the response of a rapidly changing atmospheric or ecological variable, such as fire carbon emission and burned area fraction (Supplementary Fig. 3), to a set of slowly changing environmental forcings, such as SST, soil moisture, or LAI[17,18,24–26]. At timescale, $\tau$ (currently assigned to 1 month), which exceeds the persistence time of the target fire variable, a fire variable at time $t$, $F(t)$, can be approximately decomposed into both the internal noise, $N(t)$, and the response to a set of slowly evolving variables, $\mathbf{O}(t)$, in the forcing matrix, $\mathbf{O}$, such that

$$F(t) = \mathbf{RO}(t) + N(t). \qquad (1)$$

$\mathbf{R}$ represents the response vector, which quantifies the instantaneous influence of slowly evolving oceanic and terrestrial forcings on a terrestrial flux. Multiplying the transposed forcing matrix at an earlier time, $\mathbf{O}^{\mathrm{T}}(t - \tau)$, on both sides of Eq. (1) and application of the covariance yield the following equation in covariance, $\mathbf{C}$:

$$\mathbf{C}_{FO}(t) = \mathbf{RC}_{\mathbf{OO}}(t) + \mathbf{C}_{NO}(t). \qquad (2)$$

Because oceanic and terrestrial variability cannot be forced by, or drive, subsequent ecological internal noise, $\mathbf{C}_{NO}(\tau)$ is theoretically equal to zero, allowing the feedback response vector to be estimated as

$$\mathbf{R} = \mathbf{C}_{FO}(t)\mathbf{C}_{\mathbf{OO}}^{-1}(t). \qquad (3)$$

Because the present study focuses on the seasonal timescale, the seasonal cycle and long-term linear trend are removed from the forcing and response variables prior to applying SGEFA. The present analysis is conducted for the period from 1997 to 2016, corresponding to the coverage of the Global Fire Emissions Database (GFED)[4,38]. The latest dataset, GFED4s, combines satellite information on burned area and small fire fraction with observations of vegetation productivity and meteorology to estimate gridded monthly burned area and fire emission. To increase the effective sample size, seasonal feedbacks are examined by aggregating data from three consecutive months at a time. To obtain more reliable estimates of the feedback response vector, relatively unimportant forcings are dropped from the forcing matrix to reduce the number of simultaneously considered forcings and thus minimize sampling error. This is performed via the backward-selection stepwise method[39] that optimizes the Akaike information criterion[40], an index that quantifies the quality of the statistical model by estimating the goodness of fit and penalizing based on the number of predictors.

In this study, the forcing matrix initially contains 16 oceanic forcings and two terrestrial forcings. The oceanic forcings consist of the principal component time series from the leading two empirical orthogonal functions (EOFs) of SSTs from eight basins, namely, the tropical Pacific (20 °S–20 °N, 120 °E–60 °W), tropical Atlantic (20 °S–20 °N, 70 °W–20 °E), tropical Indian (20 °S–20 °N, 35 °E–105 °E), North Pacific (20 °N–60 °N, 120 °E–100 °W), North Atlantic (20 °N–60 °N,

90 °W–10 °W), South Pacific (60 °S–20 °S, 150 °E–70 °W), South Atlantic (60 °S–20 °S, 70 °W–20 °E), and South Indian (60 °S–20 °S, 20 °E–120 °E) Oceans (as displayed in Yu et al.[17], Supplementary Fig. 7). The terrestrial forcings for each target region include the time series of the area-averaged local LAI and surface-layer (0–10 cm) soil moisture. The SGEFA forcing matrix is computed using SSTs from the Hadley Center Sea Ice and Sea-Surface Temperature dataset, LAI from three satellite-based datasets, and surface-layer soil moisture from two observation- and reanalysis-based datasets, as outlined in Supplementary Table 1. The stepwise selection and subsequent feedback response vector estimation are performed for each fixed 3-month season (January–March, FMA, … December–February). The Monte Carlo bootstrap method, with 1000 random iterations of the time series of fire activity, is applied to assess the statistical significance of the SGEFA feedback triggered by a specific SST, soil moisture, or LAI forcing, checking for 90% confidence level ($p < 0.1$).

**MLT-based prediction system of African fire.** MLTs have been used for identifying empirical regulators of fire activity. For example, the random forest (RF) method has been applied to diagnose the emergent relationships between global burned area and environmental and anthropogenic factors in both observations and dynamic global vegetation models[12]. Although MLTs lack the capability of quantifying environmental controls on fire variability[23], they provide powerful tools for building prediction systems and assessing the predictability as they account for nonlinear and interactive roles among predictors[27], which is particularly essential for fire prediction[12,22,23].

For this study, the MLT-based prediction system uses antecedent atmospheric and socioeconomic factors, as identified in previous studies[12,41,42], as well as the SGEFA-employed ocean and land surface variables. All predictors are listed in Supplementary Table 1 with their data sources, temporal coverages, and spatial resolutions. The oceanic and terrestrial predictors consist of the 16 oceanic and two terrestrial forcings used in the present SGEFA analysis. The atmospheric predictors include the occurrence of lightning and low-level atmospheric temperature, moisture, and wind speed. The socioeconomic predictors include population density and land use and land cover change. Based on data availability, the environmental predictors range from one to three months in advance of the prediction time, while the socioeconomic predictors consider only the more recent statistics typically reported at the end of the antecedent year.

In order to minimize the prediction uncertainty associated with the machine learning algorithms selected, this study examines five MLTs, including RF, support vector machine, artificial neural network, least absolute shrinkage and selection operator, and gradient boosting machine. These five algorithms differ substantially in their function. The combination of these algorithms is thus believed to better capture the complex interrelation between the forcings and response variable than any single algorithm. The 20-year data are randomly split into a 15-year training dataset and a 5-year testing dataset. The prediction model is fitted for each MLT using the training dataset, with parameters optimized for the minimum root-mean-square error via 10-fold cross-validation. The performance of the prediction model for all MLTs is evaluated using the correlation coefficient ($R^2$) between the observed and predicted time series of fire emission or burned area fraction, while reporting the highest $R^2$ among all the currently used MLTs as the predictability of fire activity. The whole model fitting and evaluation procedure is repeated for 100 random iterations of data splitting, constituting a 100-member prediction model ensemble and checking for the uncertainty of the predictability associated with interannual variability of fire activity.

The assessed models include season-specific models, in which the MLTs are built and applied by season, and the all-season models, in which data from all seasons are used in the training and testing of the MLTs. For the all-season model, we perform an additional test of the robustness of MLT-based prediction. In the additional test, we randomly split the 20-year data (240 months) into a 180-month training dataset and 60-month testing dataset, regardless of year and season, and perform the same model fitting and validation analysis. The resulting performance of all-season models fitted from the 180-month training dataset with unbalanced sampling from each season is generally worse than the performance of models fitted from the 180-month training dataset with balanced sampling from each season. This additional test further confirms that environmental controls on African fire activity are seasonally dependent.

## Data availability
The datasets utilized in this study are derived from published sources, cited in the Supplementary Table 1. The data that support the plots within this paper and other findings of this study are available from the corresponding authors upon request.

## Code availability
The code to carry out the current analyses is available from the corresponding author upon request.

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

## Acknowledgements

This research was supported by funding provided by the Environmental Sciences Division at Oak Ridge National Laboratory (ORNL), and partially supported by the Reducing Uncertainties in Biogeochemical Interactions through Synthesis and Computation Science Focus Area (RUBISCO SFA) and project under contract DE-SC0012534 funded through the Regional and Global Model Analysis activity in the Earth and Environmental Systems Sciences Division (EESSD) of the Biological and Environmental Research (BER) office in the US Department of Energy (DOE) Office of Science. This research is also partially supported by the Energy Exascale Earth System Model (E3SM) project funded through the Earth System Model Development activity in the EESSD of the BER office in the US DOE Office of Science. ORNL is managed by UT-BATTELLE, LLC, for DOE under Contract No. DE-AC05-00OR22725.

## Author contributions

J.M. conceived the research; Y.Y. and J. M. performed the analysis; and Y.Y., J.M., P.E.T., M.N., S.D.W., X.S., F.M.H., and Y.W. wrote the paper.

## Competing interests

The authors declare no competing interests.
