## [Peer Review File · Nature Communications]

Reviewers' comments:

Reviewer #1 (Remarks to the Author):

This paper uses a combination of machine learning techniques and a Stepwise Generalised Equilibrium Feedback Assessment to assess the drivers and predictability of fire over Africa on a seasonal timescale. The authors find that oceanic and terrestrial drivers are important factors in fire predictability on these timescales, not least because the standard environmental and socioeconomic drivers used in fire models have limited seasonal predictability.

This paper provides a useful and novel analysis of oceanic and terrestrial drivers of fire on a seasonal timescale, which has not been considered before to my knowledge. As the authors correctly point out, this will be interesting to those in the fire modelling community, for model development and evaluation. Fire models currently lack skill in seasonal cycles, so this could perhaps provide a way of addressing this limitation.

The method used here of combining the statistical method of SGEFA with machine learning I think is good and well-defended in the paper. The methods are clear and descriptive. I think the results on-the-whole are convincing, although in parts overly complex and difficult to follow, as outlined below. I also feel that the paper suggests that fire models only take into account climate and socio-economic predictors, but it should be noted that most do already factor in soil moisture variability, and some also account for LAI, and in fully coupled Earth System Models SSTs would also alter the climate and therefore would be accounted for. I think the paper would also benefit from discussing how these results could be used in a global model where the fire drivers are likely to be different from those over Africa. Considering many fire models in FireMIP for example are global in scale, taking account of different predictors needs to be globally applicable. If these points were addressed and the results tidied up, I think this will be a useful paper.

Line 57: The figure referred to doesn't support the text here. As the figure only shows Africa, you cannot tell from this that 50% of emissions and 70% of global burned area occurs in Africa

Line 68/69: Reword - since the 1980s

Lines 36-43: References needed

Supplementary fig 5: More information needed in the caption text. Do panels i to l also refer to ENSO, Atlantic Nino, LAI, Soil Moisture?

Fig 1: More information needed in the caption text. What are the leading two SST principal components?

Line 136: Specify what the dry fire-active season is. It says this is highlighted in Figure 1, but I can't see that it is

Line 146: Reference needed

Line 163/4: Fire active season not marked again in Figure 2

Lines 162-175: It is difficult to follow the results here, as a result of switching between the Main and the Supplementary Information, and so many panels on the figures. If the panels were referred to for specific results at least, it would make it slightly easier to follow. I wouldn't say that the influence of

the Atlantic Niño on fire carbon emissions is spatially homogenous. Perhaps less variable than ENSO. I also can't see that carbon emissions are enhanced across all of Northern Africa with El Niño, when there seem to be strong areas of decrease in central Sahara in OND, JFM, and DJF. The information on the maps is very hard to follow, not helped by very small text for the months on the map. Perhaps you might consider amalgamating the separate months into four seasons for the maps, without losing too much information.

Figure 3 is more compelling. But more information needed on what the shading, dashed lines and small dots indicate in the caption

Reviewer #2 (Remarks to the Author):

Review of paper titled "Quantifying the drivers and predictability of fire variability in Africa" by Yu et al.

General comment: accepted after major revisions

In this paper, the author applied a combined SGEFA-MLTs analytical framework to quantify the seasonal drivers and predictability of African fire variability. The authors found that ENSO, Atlantic Niño, LAI, and soil moisture have significant contributions to seasonal predictability of African fire variability. The combined SGEFA-MLT approach based on these variables and other anthropogenic drivers can be used for seasonal forecasts of African fire.

This study is meaningful, because biomass burning in Africa has strong influences on not only regional climate and hydrological cycle, but also society and economics. The analyses and presentations of the results are also straightforward. However, I do have some concerns mainly about the results and discussions.

In the responses to the drivers shown in Fig. 2, the spatial distributions of responses are quite heterogeneous, with large positive responses next to negative responses. The influences from SST on precipitation should still be large-scale, thus these heterogeneous responses should be clarified.

There is no filled circle in some seasons, e.g., OND, NDJ, DJF in Fig. 2b. What is the reason for these missing averages?

For the cross validations using the 5-year data set, the results will be more robust if the months in the 5-year data set are also randomly sampled.

I wonder why the predictability of ASO season in Northern Africa is much smaller than other seasons.

Some minor comments:

In Fig. 3 and Fig. 4, the 10% and 90% percentiles shown as shading and dash lines are not clear, especially for the shading that were overlaid. The authors should think about how to present the results clearer.

There are some typos, which need to be carefully examined and corrected, for example, page 4 line 110 "extracting" should be "extract".

Reviewers' comments:

Reviewer #1

This paper uses a combination of machine learning techniques and a Stepwise Generalised Equilibrium Feedback Assessment to assess the drivers and predictability of fire over Africa on a seasonal timescale. The authors find that oceanic and terrestrial drivers are important factors in fire predictability on these timescales, not least because the standard environmental and socioeconomic drivers used in fire models have limited seasonal predictability.

This paper provides a useful and novel analysis of oceanic and terrestrial drivers of fire on a seasonal timescale, which has not been considered before to my knowledge. As the authors correctly point out, this will be interesting to those in the fire modelling community, for model development and evaluation. Fire models currently lack skill in seasonal cycles, so this could perhaps provide a way of addressing this limitation.

Response: Thank you very much for your encouragement and suggestion. In the revised discussion section, we reflected the value of our framework in improving the simulation of fire seasonal cycles:

“Previous studies have demonstrated that the majority of FireMIP models were able to capture the spatial mean characteristics of observed global fire but failed to reproduce the seasonal to interannual variations^{16,36}. Beyond the already identified incorrect response to land use and land cover changes³⁶, potential misrepresentation of season-dependent, primary environmental drivers may induce unrealistic simulation of regional fire variability. Therefore, the SGEFA-based process quantification would provide valuable analytical tool and relationship metrics for comprehensive benchmarking of fire models.” (page 12, lines 319-326)

The method used here of combining the statistical method of SGEFA with machine learning I think is good and well-defended in the paper. The methods are clear and descriptive. I think the results on-the-whole are convincing, although in parts overly complex and difficult to follow, as outlined below. I also feel that the paper suggests that fire models only take into account climate and socio-economic predictors, but it should be noted that most do already factor in soil moisture variability, and some also account for LAI, and in fully coupled Earth System Models SSTs would also alter the climate and therefore would be accounted for.

Response: In the revised manuscript, we now more clearly point out the established knowledge and knowledge gaps from these statistical and dynamical models:

“Moreover, while different modes of variability in SSTs potentially co-impact regional fire activity in Africa²⁴, their synergetic and independent roles lack systematic exploration in either observations or model simulations.” (page 3, lines 92-95)

“Though the vegetation and soil components are parameterized in current generation of fire models⁵, their linear and nonlinear impacts on fire activity remain relatively simplified, especially on the seasonal timescale.” (page 4, lines 103-105).

I think the paper would also benefit from discussing how these results could be used in a global model where the fire drivers are likely to be different from those over Africa. Considering many fire models in FireMIP for example are global in scale, taking account of different predictors needs to be globally applicable.

Response: We now provide two potential approaches to extend our framework to global application. One approach is to apply to other fire-active regions:

“Nevertheless, efficient application of the SGEFA-MLT framework onto other fire-active regions would require careful selection of oceanic, terrestrial, atmospheric, and socioeconomic predictors that are specifically relevant to the focal region. For example, the potential influence of Madden Julian Oscillation³¹ and broader-scale vegetation anomalies³² need to be fully assessed when investigating the drivers and predictability of fire variability in Australia.” (page 11, lines 299-304)

Another approach is to apply the current framework to every pixel globally:

“Overcoming these limitations to achieve SGEFA-MLT application at fine scales would require longer observational data records with higher spatio-temporal resolution, and increased numbers of observable environmental variables.” (page 13, lines 345-347)

Line 57: The figure referred to doesn't support the text here. As the figure only shows Africa, you cannot tell from this that 50% of emissions and 70% of global burned area occurs in Africa

Response: The Supplemental Figure 1 has now included panels of (a) global fire carbon emission and (b) global burned area.

Line 68/69: Reword - since the 1980s

Response: This sentence has been changed to

“Fire activity across Africa is highly variable and has been previously attributed to changes in weather patterns, shifting plant communities, and human activity^{8,10-14}.” (page 3, lines 67-68)

Lines 36-43: References needed

Response: Nature Communications requires that the abstract not include any references.

Supplementary fig 5: More information needed in the caption text. Do panels i to l also refer to ENSO, Atlantic Niño, LAI, Soil Moisture?

Response: The caption of Supplemental Figure 5 has been changed to

“**Supplemental Figure 5:** Response of (a-d) 2-m air temperature ($^{\circ}\text{C } \sigma_{\text{forcing}}^{-1}$), (e-h) precipitation ($\text{mm day}^{-1} \sigma_{\text{forcing}}^{-1}$), and (i-l) 2-m wind speed ($\text{m s}^{-1} \sigma_{\text{forcing}}^{-1}$) to select environmental forcings in the season with corresponding maximum response in wildfire carbon emission according to SGEFA. These forcings include (a, e, i) ENSO, (b, f, j) Atlantic Niño mode, (c, g, k) LAI, and (d, h, l) soil moisture. Only statistically significant ($p < 0.1$) responses are shown here. Figure elements are the same as in Figure 2 (e-h).” (page 8 of Supplemental Materials)

Fig 1: More information needed in the caption text. What are the leading two SST principal components?

Response: The caption of Figure 1 has been changed to

“**Figure 1: Multi-dataset robustness of oceanic and terrestrial controls on African fire carbon emission by season according to SGEFA.** Color represents the number of dataset combinations [out of the currently examined six dataset combinations (Supplemental Table 1)] that indicate significant ($p < 0.1$) response of regional average fire carbon emission to either of the leading two principal components' (PCs) time series corresponding to the SST Empirical

Orthogonal Functions (EOFs) from eight oceanic basins [North Atlantic (NA), North Pacific (NP), tropical Indian (TI), tropical Atlantic (TA), tropical Pacific (TP), South Indian (SI), South Atlantic (SA), and South Pacific (SP)], and regional average LAI and 0-10 cm soil moisture (SM), across (a) northern Africa and (b) southern Africa. The fire-active season is highlighted with red x-axis labels. The forcings are detailed in the Methods section.”

Line 136: Specify what the dry fire-active season is. It says this is highlighted in Figure 1, but I can't see that it is

Response: The dry and wet seasons are specified in the main text:

“Northern Africa's fire is sensitive to variability in tropical ocean SSTs, especially those from the tropical Atlantic Ocean during the dry season (November to March) and the tropical Indian Ocean during the wet season (April to September). During the boreal winter, North Atlantic Ocean SSTs play a substantial role in regulating the northern African fire carbon emissions (Figure 1a). SSTs from Southern Hemispheric oceans, especially the South Atlantic Ocean, exert relatively strong control compared with the tropical and Northern Hemispheric oceans on southern African regional fire during the fire-active season (May to November) (Figure 1b).”
(page 5, lines 140-148)

We also changed the highlight to red x-axis labels to indicate the fire-active season, as noted on the revised caption of Figures 1, 2, and 4.

Line 146: Reference needed

Response: This sentence has been revised:

“In terms of terrestrial drivers of African fire, the observed effects from soil moisture changes are more robust than LAI changes, especially during the wet season in northern Africa (Figure 1a).” (page 5, lines 150-152)

Line 163/4: Fire active season not marked again in Figure 2

Response: We marked fire-active season in the revised Figure 2.

Lines 162-175: It is difficult to follow the results here, as a result of switching between the Main and the Supplementary Information, and so many panels on the figures. If the panels were referred to for specific results at least, it would make it slightly easier to follow. I wouldn't say that the influence of the Atlantic Niño on fire carbon emissions is spatially homogenous. Perhaps less variable than ENSO. I also can't see that carbon emissions are enhanced across all of Northern Africa with El Niño, when there seem to be strong areas of decrease in central Sahara in OND, JFM, and DJF. The information on the maps is very hard to follow, not helped by very small text for the months on the map. Perhaps you might consider amalgamating the separate months into four seasons for the maps, without losing too much information.

Response: Thank you for the valuable suggestion on results presentation.

Regarding the influence of Atlantic Niño mode, we changed the corresponding sentence to:

“The SGEFA-based observational analysis identifies the spatially heterogeneous influence of ENSO and relatively homogeneous influence of the Atlantic Niño mode on African fire activity, with the response magnitude of both oceanic drivers peaking in the fire-active season (panels a, b, e, f, i, j in Figure 2 and Supplemental Figure 4).” (page 6, lines 167-170)

We revised the sentence about ENSO's impacts:

“Across a large portion of northern Africa, especially the northern Sahel and West African monsoon regions, El Niño typically supports enhanced fire carbon emissions and expanded burned area during the fire-active season in boreal winter (panels a and e in Figure 2 and Supplemental Figure 4). The increased fire activity in northern and western Sahel is largely associated with El Niño-induced anomalous low-level warming and consequential fuel drying (Supplemental Figure 5a, e). Similar meteorological responses to El Niño (Supplemental Figure 5a, e) also support enhanced fire emissions and expanded burned area across southwestern Africa during the fire-active season in boreal summer (panels e and i in Figure 2 and Supplemental Figure 4).” (page 6, lines 170-179)

We changed Figure 2 for offering a clearer presentation.

Furthermore, we now refer to specific panels of each figure throughout the revised manuscript.

For example, the discussion about terrestrial influence on fire activity is revised as:

“The SGEFA analysis uncovers generally negative responses in African fire to positive anomalies in soil moisture and complex responses to LAI changes, with greater response magnitudes to soil moisture anomalies during the fire-active season (panels c, g, k, d, h, l in Figure 2 and Supplemental Figure 4). Across most of Africa, wetter soils inhibit biomass burning (panels d, h, l in Figure 2 and Supplemental Figure 4) through surface low-level cooling and elevated amounts of precipitation (Supplemental Figure 5d, h). However, the influence of LAI on fire activity in Africa is spatially heterogeneous (panel g in Figure 2 and Supplemental Figure 4). Positive anomalies in LAI indicate a higher amount of available fuel for biomass burning, leading to enhanced fire activity over portions of the West African monsoon region and grasslands in southern Africa (panel g in Figure 2 and Supplemental Figure 4). Similarly, positive anomalies in LAI cause an overall increase in North African regional-average fire emission and burned fraction during the fire-active season in boreal winter and spring (panel c in Figure 2 and Supplemental Figure 4). However, surface cooling and decreased near-surface wind speed associated with positive LAI anomalies (Supplemental Figure 5c, k) provide unfavorable meteorological conditions for biomass burning, thereby inhibiting fire activity across the majority of southern Africa during the fire-active season in boreal summer (panels g, k in Figure 2 and Supplemental Figure 4).” (page 7, lines 188-205)

With these changes, we hope the description of results is now easier to follow.

Figure 3 is more compelling. But more information needed on what the shading, dashed lines and small dots indicate in the caption

Response: Figure 3 is revised according to Reviewer 2's comments. We revised the caption to better describe the figure elements:

“Figure 3: Predictability of the African fire carbon emission anomalies as a function of lead time, estimated using multiple MLTs. The predictability is represented by the squared correlation coefficient (R^2) between the predicted and observed monthly anomalies ($n = 60$) of the regional average fire carbon emissions across the (a) northern and (b) southern African ecoregions. The assessed sets of predictors include previously identified atmospheric and socioeconomic predictors (blue), currently identified oceanic and terrestrial predictors (black), and the combination of all these predictors (red). The assessed models include season-specific models (filled circles), in which the MLTs are built and applied by season, and all-season models (open squares). The circles and squares indicate the mean R^2 across the 100 ensemble members

of the best MLT (Methods), and the vertical lines indicate the range of 10th and 90th percentiles of the 100 ensemble members.”

Reviewer #2

General comment: accepted after major revisions

In this paper, the author applied a combined SGEFA-MLTs analytical framework to quantify the seasonal drivers and predictability of African fire variability. The authors found that ENSO, Atlantic Nino, LAI, and soil moisture have significant contributions to seasonal predictability of African fire variability. The combined SGEFA-MLT approach based on these variables and other anthropogenic drivers can be used for seasonal forecasts of African fire.

This study is meaningful, because biomass burning in Africa has strong influences on not only regional climate and hydrological cycle, but also society and economics. The analyses and presentations of the results are also straightforward.

We truly appreciate the positive comments.

However, I do have some concerns mainly about the results and discussions.

In the responses to the drivers shown in Fig. 2, the spatial distributions of responses are quite heterogeneous, with large positive responses next to negative responses. The influences from SST on precipitation should still be large-scale, thus these heterogeneous responses should be clarified.

Response: In the revised manuscript, we attributed the heterogeneous responses in Figure 2 to the season-dependent response:

“The season-dependent response in fire to oceanic drivers (Figure 2a-b, i-j) partly leads to the spatial heterogeneity of the seasonal maximum-magnitude responses presented in panels e-f of Figure 2 and supplemental Figure 4.” (page 7, lines 185-187)

The seasonally dependent and spatially heterogeneous response to oceanic and terrestrial drivers is further discussed in the discussion section:

“The SGEFA analysis identifies spatially heterogeneous and seasonally dependent influences of key oceanic and terrestrial forces on African fire activity. The currently identified, spatially heterogeneous ENSO signal on the African fire carbon emission and burned fraction is largely consistent with previous conclusions based on regression and correlation^{10,22}. Beyond ENSO, this study further demonstrates the observed role of tropical Atlantic Ocean SSTs on African fire activity, through the SSTs’ influence on the climate of Sahel and West African monsoon regions as noted in previous studies³³. Compared to LAI, the observational analysis further exhibits more robust effects on fire from changes in soil moisture. Such stronger role of soil moisture on African fire activity is supported by previous modeling and observational studies³⁴. They concluded that the Sahel ecoregion represents one of the global hotspots of soil moisture-atmosphere coupling, with direct soil moisture feedbacks outweighing the influence of vegetation²⁰.” (page 12, lines 305-316)

There is no filled circle in some seasons, e.g., OND, NDJ, DJF in Fig. 2b. What is the reason for these missing averages?

Response: We added an explanation of the missing filled circles in the caption of revised Figure

2, read as “A missing filled circle in (a-d) and (i-l) indicates insignificant multi-dataset average response to the specific forcing.”

For the cross validations using the 5-year data set, the results will be more robust if the months in the 5-year data set are also randomly sampled.

Response: Thanks for the insightful suggestion. We performed the recommended robustness test, and briefly discussed it:

“The assessed models include season-specific models, in which the MLTs are built and applied by season, and the all-season models, in which data from all seasons are used in the MLTs’ training and testing. For the all-season model, we perform an additional test of the robustness of MLT-based prediction. In the additional test, we randomly split the 20-year data (240 months) into a 180-month training dataset and 60-month testing dataset, regardless of year and season, and perform the same model fitting and validation analysis. The resulting performance of all-season models fitted from the 180-month training dataset with unbalanced sampling from each season is generally worse than the performance of models fitted from the 180-month training dataset with balanced sampling from each season. Such additional test further confirms the environmental controls on African fire activity are seasonally dependent.” (page 16, lines 450-460)

I wonder why the predictability of ASO season in Northern Africa is much smaller than other seasons.

Response: A hypothesis regarding the lower predictability of fire activity in Northern Africa in ASO is provided in the revised manuscript:

“However, in ASO, northern Africa shows relatively weak fire predictability, likely caused by reduced impacts of broad-scale atmospheric circulation from key ocean-atmosphere teleconnection patterns²⁸, including ENSO and Atlantic Niño mode (Figure 1a and Figure 2a, b).” (page 10, lines 267-270)

Some minor comments:

In Fig. 3 and Fig. 4, the 10% and 90% percentiles shown as shading and dash lines are not clear, especially for the shading that were overlaid. The authors should think about how to present the results clearer.

Response: We revised Figure 3 and Figure 4 for clearer presentation.

There are some typos, which need to be carefully examined and corrected, for example, page 4 line 110 “extracting” should be “extract”.

Response: The sentence has been changed to:

“Main challenges to extracting key oceanic and terrestrial drivers of African regional fire activity include the following...”. (page 4, line 113)

We also examined the entire manuscript and corrected for any other typos.

Reviewer #1 (Remarks to the Author):

The authors have addressed all of the review comments made, in my opinion satisfactorily. The text is now clearer with specific references to figures in the Results section, and the figures are improved with 10th and 90th percentiles now shown with vertical lines in figure 3 and 4, and clearer figure captions.

Overall, the usefulness of this paper is in assessing the seasonal predictability of fire activity in Africa 1-6 months ahead, and in improved understanding of the environmental drivers. The approach seems novel, and would be of interest to other researchers in the field. I believe the methods are sound, and are well-described, and the figures and results give evidence to support the conclusions. I wouldn't say the results would be of very wide interest outside of the discipline, as the results are quite specific to one region and its drivers, but I think the study is useful and interesting.

Reviewer #2 (Remarks to the Author):

The authors have well addressed my concerns. Hence, I recommend acceptance for publication.

Dear reviewers,

Thank you both for the positive feedback on our revision. We greatly appreciate your constructive comments and suggestions during the review.

Reviewer #1 (Remarks to the Author):

The authors have addressed all of the review comments made, in my opinion satisfactorily. The text is now clearer with specific references to figures in the Results section, and the figures are improved with 10th and 90th percentiles now shown with vertical lines in figure 3 and 4, and clearer figure captions.

Overall, the usefulness of this paper is in assessing the seasonal predictability of fire activity in Africa 1-6 months ahead, and in improved understanding of the environmental drivers. The approach seems novel, and would be of interest to other researchers in the field. I believe the methods are sound, and are well-described, and the figures and results give evidence to support the conclusions. I wouldn't say the results would be of very wide interest outside of the discipline, as the results are quite specific to one region and its drivers, but I think the study is useful and interesting.

Reviewer #2 (Remarks to the Author):

The authors have well addressed my concerns. Hence, I recommend acceptance for publication.